# Clinical Features of Human Parvovirus B19-Associated Encephalitis Identified in the Dakar Region, Senegal, and Viral Genome Characterization

**DOI:** 10.3390/v17010111

**Published:** 2025-01-15

**Authors:** Al Ousseynou Seye, Fatou Kiné Top, Maimouna Mbanne, Moussa Moise Diagne, Ousmane Faye, Amadou Alpha Sall, Ndongo Dia, Jean-Michel Heraud, Martin Faye

**Affiliations:** Virology Department, Institut Pasteur de Dakar, 36 Avenue Pasteur, Dakar 200, Senegal; alousseynou.seye-ext@pasteur.sn (A.O.S.); fatoukine.top@pasteur.sn (F.K.T.); maimouna.mbanne@pasteur.sn (M.M.); moussamoise.diagne@pasteur.sn (M.M.D.); ousmane.faye@pasteur.sn (O.F.); amadou.sall@pasteur.sn (A.A.S.); ndongo.dia@pasteur.sn (N.D.); heraudj@who.int (J.-M.H.)

**Keywords:** human parvovirus B19, encephalitis, PCR, sequencing, Senegal

## Abstract

Neurological manifestations associated with human parvovirus B19 (B19V) infections are rare and varied. Acute encephalitis and encephalopathy are the most common, accounting for 38.8% of all neurological manifestations associated with human B19V. Herein, we report on the clinical features of 13 laboratory-confirmed human cases of B19V-associated encephalitis in Senegal in the framework of a hospital-based surveillance of acute viral encephalitis conducted from 2021 to 2023. Overall, B19V was detected from 13 cerebrospinal fluid samples using specific real time PCR. The mean age was 16.7 years among B19V-positive patients, with a higher prevalence in 0–5-year-old children and the sex ratio (male/female) was 2.25. The B19V-positive patients mainly exhibited hypoleukocytosis, normal glycorrhachia, and normal proteinorrachia in the cerebrospinal fluid. While the main neurological symptoms included meningeal and infectious syndromes. Furthermore, three complete B19V genome sequences were successfully characterized using next-generation sequencing. The newly characterized sequences belonged to the genotype 1a and represent, to date, the first complete B19V genome sequences from Senegal. These sequences could be useful not only in future phylodynamic studies of B19V but also in the development of prevention or treatment countermeasures. Our study is noteworthy for the identification of acute B19V-associated encephalitis in Senegal More investigations on the risk factors associated with B19V transmission in Africa are warranted.

## 1. Introduction

Human parvovirus B19 (B19V), first identified in asymptomatic blood donors during hepatitis B screening [1], spreads easily through respiratory droplets and causes erythema infections across all age groups [2]. B19V infections are generally benign, self-limiting, and are controlled by the development of a specific immune response, but in many cases, clinical situations can be more complex and require therapy. However, B19V infection can lead to adverse health outcomes among people without pre-existing immunity who are pregnant, immunocompromised, or have chronic hemolytic disorders. B19V can also spread through transfusions and contaminated blood products. Although B19V infections often resolve spontaneously without adverse outcomes, the risk of an adverse fetal outcome is 5–10% and is higher when acute infection occurs between gestational weeks 9–20. B19V infection may cause chronic or temporary aplastic anemia and, in immunocompromised patients, can result in issues like glomerulonephritis, myocarditis, liver failure, or neurological complications. B19V infection occurs worldwide, with epidemics tending to follow a 3–6-year cycle, while seroprevalence rates vary considerably by age and geographical area. The lowest seroprevalence rates were reported in pre-school children (15%), while the highest were found in adults, reaching up to 85% in the elderly. Particularly, in Africa, IgG seroprevalence rates in pregnant women vary from 20% to 82% [3].

Currently available treatments, such as intravenous immunoglobulin administration, are only supportive and non-specific and are often of limited efficacy [4].

B19V is a member of the *Parvoviridae* family and the *Erythroparvovirus* genus that interacts with the blood group P antigen (globoside) [5,6]. The genome of B19V is a single-stranded deoxyribonucleic acid (ssDNA) molecule of either polarity, 5596 nucleotides (nt) long, composed of two terminal regions, 383 nt, that provide the origins of replication, flanking a unique internal region, 4830 nt, containing all open reading frames. The viral genome presents two major ORFs in its internal region, in the left side for the non-structural protein (NS) and in the right side for the two colinear capsid proteins, VP1 and VP2 [4]. B19V presents genetic diversity with three genotypes (1–3). Genotype 1 is circulating worldwide [7] and has replaced, over the last fifty years, genotype 2 [8], which is now found sporadically. Genotype 3 can be found at a lower frequency in restricted geographic areas [7]. Nucleotide distances between the three genotypes are up to 10%. However, the intragenotype nucleotide distances are lower. They range between 3 and 8% for genotypes 2 and 3 and between 1 and 3% for genotype 1, possibly reflecting a shorter evolutionary history for the latter [9]. Genotypes 1 and 3 are further categorized into subtypes a and b [10].

B19V infection has been previously associated with a variety of neurological complications, including encephalitis, meningitis, stroke, neuropathy, status epilepticus, and encephalopathy [11,12,13,14]. Of these, encephalitis and encephalopathy are the most common, accounting for 38.8% of all B19V-associated neurological complications [14]. The last 10 years have witnessed a wave of case reports of B19V-associated encephalitis [15,16,17,18], and the majority of cases included children [19].

However, there is a scarcity of data on the molecular characterization and analysis of the viral genome identified from cerebrospinal fluid (CSF) samples [20], particularly in Africa, where resources are limited. However, a study conducted in Egypt between 2016 and 2019 identified the presence of B19V DNA in 30 patients hospitalized for neurological complications, with HEV, EBV, or VZV coinfections [21].

The diagnosis of B19V infection is based on the detection of either B19V DNA by polymerase chain reaction (PCR) or specific antibodies against B19V in serum or CSF samples [13]. Next-generation metagenomic sequencing (mNGS) technology is a high-throughput approach that is increasingly used for the identification of all the genetic materials present in a biological sample and has demonstrated its usefulness to complete standard diagnostic methods [22].

Herein, we report on clinical features of B19V-associated infection identified from patients with acute encephalitis, hospitalized in two referral hospitals in the Dakar region in Senegal and the characterization of the entire B19V genome sequences from CSF samples using molecular techniques and mNGS.

## 2. Materials and Methods

### 2.1. Ethical Statement

This research project has been approved by the National Ethics Committee for Health Research of the Ministry of Health and Social Action (authorization no. 0094/MSAS/CNERS/SP, dated 19 April 2022). Specimens were collected as part of routine clinical care procedures for the management of patients presenting with encephalitis. Lumbar punctures were performed by experienced clinicians if recommended for the diagnosis of the cause of encephalitis and to adapt the treatment consequently. All virology tests were performed at no cost for the patient. A unique identification number (ID) was assigned for each sample collected to anonymize data from included patients.

### 2.2. Study Design

Through the research project entitled “Surveillance of infectious encephalitis in Senegal (ENSENE)”, a prospective longitudinal study was conducted from April 2021 to December 2023. Patients presenting clinical features of encephalitis [23] were recruited from two neurology departments in the Dakar region, Senegal (Fann National University Hospital and Albert Royer National Hospital).

The study defines suspected encephalitis in adults (>15 years) as altered mental status for over 24 h without another cause, along with at least two other criteria such as fever, seizures, or specific imaging/CSF findings. For children (≤15 years), it involves acute meningoencephalitis with at least one sign like fever, convulsions, or neurological deficits. Inclusion requires meeting the case definition, having a symptom duration under 10 days, and consent for lumbar puncture, while exclusion covers contraindications or failure to meet criteria. Non-inclusion applies to alternative diagnoses such as brain abscess or metabolic encephalopathies.

Samples consisted of CSF obtained by lumbar puncture, nasopharyngeal swabs and blood and were accompanied by socio-demographic information sheets. Samples were sent the same day to the medical laboratory and to the national reference laboratory for rabies and viral encephalitis at Institut Pasteur de Dakar (IPD). Viral encephalitis is confirmed by virus detection in CSF and an etiology is considered as probable if detected in other sample types.

### 2.3. Biological Testing

Nasopharyngeal swabs were screened for the detection of viral respiratory viruses, while blood was used for serology testing targeting antibodies against arboviruses.

CSF was analyzed for cytology, biochemistry, bacteriology, parasitology, and mycology. Briefly, viral nucleic acid (ribonucleic acid (RNA) and DNA) was extracted from 200 µL of CSF using the Veri-Q PREP M16 nucleic acid extraction system (MiCoBioMed, Seongnam-si, Republic of Korea), according to the manufacturer’s instructions. The nucleic acid was eluted in 60 μL of nuclease-free water and stored at −80 °C until analysis. Nucleic acids were assayed by Allplex Meningitis-V1 and -V2 IVD kits (Seegene Inc., Seoul, Republic of Korea). The DNA extracts were tested for the presence of herpes simplex virus 1 (HSV1), herpes simplex virus 2 (HSV2), varicella zoster virus (VZV), Epstein-Barr virus (EBV), cytomegalovirus (CMV), human herpesvirus 6 (HHV6), human herpesvirus 7 (HHV7), adenovirus (AdV), and B19V using real-time PCR, while the RNA extracts were analyzed for the presence of enterovirus (HEV), human parechovirus (HPeV), mumps virus (MV) by real-time reverse transcription (RT)-PCR. All experiments were performed on a CFX96 Dx equipment for in vitro diagnostics (IVDs) (Bio-Rad, Singapore) and data were then analyzed using the IVD program Seegene Viewer for Real time Instruments version 3.24.000 (Seegene Inc., Seoul, Republic of Korea).

### 2.4. B19V Sequencing

The DNA was quantified by a Qubit 1X dsDNA HS Assay kit version Q33231 (Thermo Fisher Scientific, Waltham, MA, USA) and 10 ng of DNA was used with the TWIST EF 2.0 sequencing protocol version DOC-001170 REV 4.0 (Twist Bioscience, South San Francisco, CA USA). The purified DNA was fragmented into 400 bp segments, and the libraries were prepared using the TWIST Universal Adapter System-Truseq version 101308 (Twist Bioscience, USA). The indexed libraries were purified using magnetic beads and amplified by PCR. After washing with magnetic beads, the libraries were quantified using Qubit and pooled into groups of 8 samples. The pooled libraries were enriched using two mixtures, including one with the enrichment probes and another with the blockers. After heating and cooling, the hybridization reaction was incubated at 70 °C for 16 h. The beads attaching the enriched libraries were successively subjected to three wash-hybridization cycles using a binding buffer. The enriched libraries were then amplified by PCR using the equinox amplification mix and primers. The purified libraries were then quantified and normalized to 2 nM. The experiment was performed on the Illumina Miseq instrument, and consensus genomes were generated using Chan Zuckerberg ID (https://czid.org/, accessed on 2 December 2024).

### 2.5. Genetic Analysis of the B19V Sequences

The sequences obtained were meticulously aligned and manually curated using the Alignment Program MAFFT version 7 (https://mafft.cbrc.jp/alignment/server/, accessed on 2 December 2024) and the software BioEdit version 7.2.5 [24], which uses the W-clustal algorithm [25], respectively.

#### 2.5.1. Genetic Variability

We then determined the genetic diversity parameters (the rate of mutations (R); genetic distance between populations) of the B19V sequences obtained in this study in relation to the reference (AF162273.1) and B19V sequences downloaded from GenBank (https://www.ncbi.nlm.nih.gov/genbank/, accessed on 2 December 2024). These parameters were determined using Mega7 software version 7.0.14 [26], and the frequencies of amino acids were determined at the best reading frame (no stop codon).

#### 2.5.2. Phylogenetic Analysis

Our dataset included B19V sequences retrieved from GenBank and the newly characterized sequences from patients with encephalitis in Senegal. After multiple alignments of our dataset, a Bayesian analysis was performed to determine the most appropriate nucleotide substitution model, considering the lowest BIC (Bayesian information criterion) score. The maximum likelihood (ML) phylogenetic tree was inferred from 1000 replications using the IQTree online software (http://iqtree.cibiv.univie.ac.at/, accessed on 2 December 2024) with the best-fitted nucleotide substitution model to our dataset. The ML tree was rooted on the midpoint and nodes were marked with bootstrap values.

#### 2.5.3. Assessment of Potential Amino Acid Motifs of Virulence

To assess the presence of new mutations, our dataset of B19V sequences was screened for motives of virulence. The pathogenicity of the variants was determined for the exonic region in relation to non-synonymous mutations using Scale-Invariant Feature Transform (SIFT) software (https://sift.bii.a-star.edu.sg/, accessed on 2 December 2024). This software is a probabilistic classifier that calculates the functional significance of an allele change from the reference sequence using the amino acid sequences obtained from the Universal Protein Resource Knowledgebase (UNIPROTKB) database (https://www.uniprot.org/, accessed on 2 December 2024) Any mutation with a score greater than or equal to 0.05 is classified as a tolerated mutation.

#### 2.5.4. Data Analysis

The mean age and its standard deviation were calculated using RStudio software version 4.4.0. Tables and figures were generated using the Microsoft Excel 2019 program.

## 3. Results

### 3.1. Characteristics of Infected Patients with B19V

From April 2021 to December 2023, a total of 1416 patients were enrolled, and samples were screened for panels of viral etiologies using molecular techniques. Etiologies including herpesviruses, enteroviruses, and respiratory viruses were found. However, a total of 13 patients out of the included patients tested positive for B19V DNA by PCR in CSF samples (Appendix A). The male/female sex ratio was 9/4 (2.25). The mean age of B19V-positive patients was 16.7 years [2–53] with a standard deviation of 17.9 years, and 72.7% (8/11) had fever. In addition, there were more acute B19V infections in the 0–5-year-old age group. On examination of the CSF biological parameters, 75% (6/8) contained less than five leukocytes per µL and 25% (2/8) contained less than 50% lymphocytes. A biochemical analysis of CSF samples showed that 87.5% (7/8) of B19V DNA positives had normal glycorrhachia and normal CSF proteinorrachia (missing data for five patients). However, two out of the 13 patients with B19V-associated encephalitis had a hyperpleocytosis of more than 5 cells/µL (25%, 2/8), and one of these two patients showed a hyperproteinorrachia (>1g/L). The main neurological symptoms observed in the B19V-positive patients included meningeal syndrome (27.3%), infectious syndrome (27.3%), behavioral disorders/delusions (18.2%), progressive motor deficit and stroke (9.1%), tonic and febrile convulsions (9.1%), flaccid deficit of both lower limbs (9.1%), status epilepticus preceded by diarrhea (9.1%), progressive motor deficit of all four limbs (9.1%), and convulsive seizures and dyspnea (9.1%) (with two missing data points) (Table 1).

Interestingly, only one of the 13 patients infected with B19V DNA was coinfected with HSV1 DNA in CSF by PCR.

### 3.2. Phylogenetic Analysis

Three complete B19V sequences were successively generated by mNGS sequencing of the positive cerebrospinal fluid samples and submitted to the NCBI GenBank under the accession numbers PQ390712- PQ390714 (Appendix A). The ML phylogenetic tree was inferred using IQtree online software with the best nucleotide substitution model to our dataset. The newly characterized genomes from Senegal were grouped into the B19V genotype 1a and clustered with sequences from Argentina (AR), the Netherlands (NL), France (FR), Brazil (BR), the United States (US), Switzerland (CH), the United Kingdom (GB), and India (IN) (Figure 1).

### 3.3. Genetic Diversity and Polymorphisms

In this study, a total of three whole-genome sequences of B19V isolates from Senegal were obtained by sequencing using Twist EF 2.0. The genetic diversity of these sequences compared to different B19V sequences downloaded from GenBank was described using genetic diversity parameters obtained with MEGA 7.0.14 software. An analysis of the genetic diversity parameters showed a mutation rate R equal to 3.15 compared with the reference (AF162273.1) and a genetic distance D equal to 0.011 ± 0.002 compared with B19V genotype 1 from GenBank.

Ten new mutations were identified in B19V sequences from Senegal in comparison with the neurotropic B19V genotype 1 from Brazil (GenBank accession no. JX267259.1, JX559657.1, JX559663.1) and B19V genotypes 1 downloaded from GenBank. Of these, four (p. Ile 181 Met, p. Val 30 Leu, p. Ser 98 Asn, p. His 100 Tyr) are known to affect the function of the non-structural NS1 and VP1 capsid proteins (Table 2).

## 4. Discussion

B19V-associated encephalitis is rare but can occur in humans, particularly in children, even in the absence of B19V-typical symptoms. Herein, we present the clinical features and genomic characteristics of B19V infections identified from 13 patients with encephalitis as part of a prospective surveillance of viral encephalitis in Senegal. Clinical symptoms varied among patients, except for fever, which was the most prevalent.

B19V has been previously identified as responsible for encephalitis in 39% of reported cases, with 86% among cases with encephalitis involving children [16]. A confirmed case of B19V-associated encephalitis has also been reported in China in a seventeen-year-old boy who presented with confusion, persistent fever, and focal neurological signs, providing evidence for the implication of B19V as causative pathogen of encephalitis in humans [27].

Our data exhibited a male/female sex ratio of 2.25 with a mean age of 16.7 years [2–53], and more cases involved children aged 0 to 5 years. The gender difference observed in our study is similar to that previously reported in the USA in 2009 with a sex ratio of 1.4 [13]. In 2015, another previous study from the USA also reported 34 cases of B19V encephalitis or encephalopathy in children aged 1–15 years (mean 6.7 years; median 7 years) with a sex ratio of 0.79 [12]. These data may suggest that children of all sexes are at greater risk of developing B19V encephalitis. A previous study conducted in West Bengal, India, from 2016 to 2018 identified 13 B19V-postive cases of acute meningoencephalitis out of 403 (3.2%), also with a high prevalence in children [20]. B19V infects 1–5% of pregnant women, usually with a normal pregnancy [3,28]. Despite its highest transmissibility in respiratory droplets, vertical transmission of B19V across the placenta could have played a pivotal role in the high prevalence in children. Although the risk of fetal complications depends largely on gestational age at the time of maternal B19V infection, fetal anomalies associated with B19V are rare. However, in some rare cases, the infection could lead to damage in various fetal organs, including the brain, and cause neurological complications [22]. Thus, epidemiological pregnancy studies are needed in Africa, where resources are limited, to assess the disease’s burden in vulnerable groups such as pregnant women and newborns [29].

Interestingly, two out of eight patients (13 patients including five that were missing data) with B19V-associated encephalitis had a pleocytosis of more than 5 cells/mm^3^ (25%, 2/8), and one of these two patients showed a high protein concentration (>1 g/L), highlighting the importance of hyperleukocytosis and hyperproteinorrachia, previously described as biomarkers of viral encephalitis [27,30]. Although some patients from our study did not meet the case definition of encephalitis (e.g., no pleocytosis nor normal CSF parameters), the detection of B19V DNA in the CSF of 13 patients admitted with acute neurological signs and the absence of other well-known etiologies could suggest that B19V could be responsible for this encephalitis syndrome, as previously described [20,31]. In addition, the correction of CSF parameters could result from the admission of patients at late stages of the infection and the compensatory treatment initiated by clinicians since the first consultations in primary healthcare settings. However, similar longitudinal studies focusing on B19V infection in patients with encephalitis could be promoted in many countries in Africa for a better characterization of B19V-associated encephalitis.

The use of an mNGS sequencing technique adapted to the search for viral pathogens in infectious diseases enabled us to generate three complete B19V sequences from CSF samples. Previously applied to B19V’s genomic characterization [27,32], NGS sequencing offers advantages over conventional diagnostic methods for encephalitis to enlarge the spectrum of detection of etiologies probably involved in encephalitis syndrome. It could be useful to consider using NGS for future studies focusing on the specific identification of etiologies of encephalitis in humans. The three newly characterized B19V genomes revealed an R mutation rate of 3.15 compared to the reference (AF162273.1), a genetic distance D equal to 0.011 ± 0.002 compared with the B19V sequences from GenBank, and the presence of ten novel substitutions. Interestingly, four out of these ten mutations (Ile-181-Met, Val-30-Leu, Ser-98-Asn, His-100-Tyr) exhibited a potential of affecting the function of the non-structural NS1 and capsid VP1 proteins. The B19V non-structural proteins are essential for virus replication and packaging, playing many roles in the virus life cycle. Due to a lack of structural information on B19 NS1, the role of certain residues such as 181 in NS1 function is unclear. In one report, examining a conserved residue identified by Abe and colleagues [33], the substitution of isoleucine 181 for methionine, as occurs in B19 genotype II, results in the reduction in B19 NS1-induced cytotoxicity of liver cells. This could lead to a reduction in hepatocyte loss and thus a reduction in the B19-related liver disease burden [34]. The viral structural proteins (VP1/VP2) form the viral capsids for viral DNA encapsidation [35]. In VP1 regions covering amino acids 28 to 136 and 189 to 247, the number of amino acid exchanges is high. Conservative exchanges such as Val 30 Leu and non-conservative exchanges such as Ser 98 Asn were observed. The majority of non-conservative exchanges were observed in isolates obtained from patients chronically infected with B19V. This may indicate that prolonged viremia may be associated with a high degree of amino acid variation [36]. Studies focusing on specific mutations such as those observed in VP1 remain limited, but genetic variability in the VP1 and VP2 capsid proteins has been explored, particularly for its impact on infectivity and immune response. Research on the unique region of VP1 (VP1u) has shown that variations in this zone can affect the virus’ interactions with host cells and influence its immunogenic properties [37]. The N-terminal region of VP1u contains a cluster of epitopes recognized by neutralizing antibodies, highlighting its critical role in viral infection. While VP1u is inaccessible to antibodies in virions circulating in the bloodstream, it becomes exposed upon interaction with target cells, a prerequisite for virus uptake. This region includes a receptor-binding domain (RBD) that facilitates virus entry by binding to an unidentified receptor (VP1uR) [6]. However, further experimental studies assessing the role of these mutations on virus fitness could be needed.

The ML phylogenetic analysis showed that the three newly characterized B19V sequences from Senegal belonged to genotype 1a and clustered with previous sequences from abroad, suggesting a possible virus importation. However, there is a crucial need of generating more complete B19V genomes from Africa to better understand the probable genetic factors driving the virus presence in the CSF or the severity of the clinical manifestations, including encephalitis [19]. Genotype 1 B19V is more prevalent worldwide [20]. These newly characterized sequences from Senegal could also be useful for future phylodynamic studies and updating the existing diagnostic tools.

More investigations on the extrinsic risk factors driving B19V-associated encephalitis such as the possible role of host genetic or environmental factors in B19V pathogenesis in the central nervous system (CNS) are warranted for therapeutic guidance. To this end, B19V infection must be considered in the differential diagnosis of neurological symptoms. In addition, understanding its geographical spread and clinical impact is crucial for effective prevention and management strategies. Longitudinal studies in Africa with a large sample size and including serology [38] are also needed to better understand the risk of B19V transmission such as blood transfusion [39]. The gap in the development of antiviral strategies and particularly the lack of specific antiviral drugs is striking. New candidate vaccines are currently being developed to prevent B19V. One of these candidates is a yeast-derived VLP (virus-like particle) vaccine that has proven effectiveness in vivo and in vitro [40], inducing neutralizing antibodies against B19V. In addition, B19 VLPs formulated with MF59 adjuvant led to the generation of neutralizing antibodies against B19V in a Phase I clinical trial, suggesting that these B19V VLPs may be powerful vaccine candidates. However, inadvertent complications such as rash were reported after VLP B19 vaccination, leading to the discontinuation of this clinical trial [41].

The implementation of recommended infection control measures for patients with B19V infection in healthcare settings such as practicing good hand hygiene, avoiding sharing food or drinks, and wearing a mask or respirator while at work, could also be considered.

## Figures and Tables

**Figure 1 viruses-17-00111-f001:**
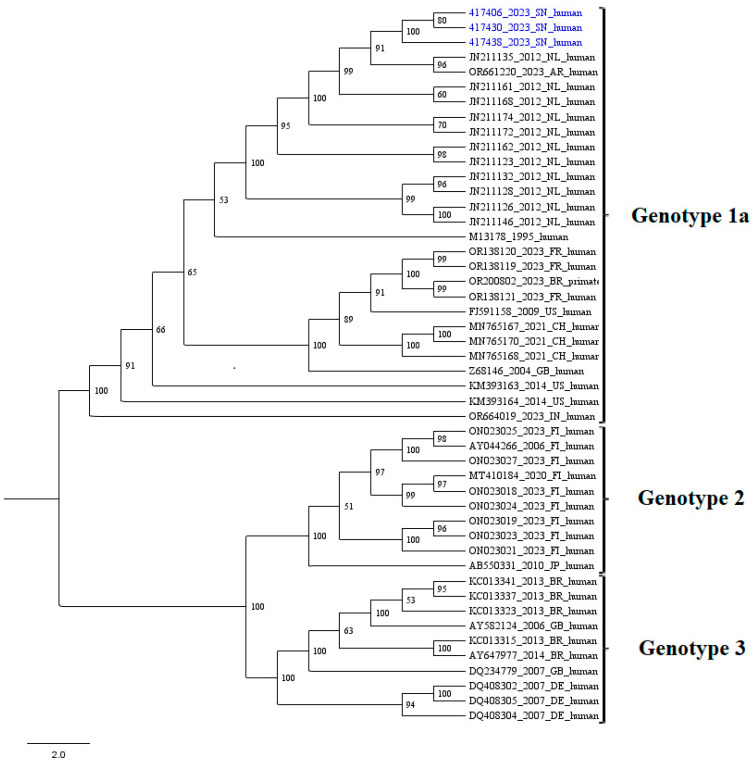
Maximum likelihood tree based on complete B19V sequences (around 5596 nt). The nodes are labeled with bootstrap values and the names of the B19V isolates are color-coded as follows: in blue, the B19V genotype 1a isolated from cerebrospinal fluid samples from Senegal; in black, the B19V genotype 1, 2, and 3 downloaded from NCBI GenBank (https://www.ncbi.nlm.nih.gov/genbank/about/, accessed on 2 December 2024).

**Table 1 viruses-17-00111-t001:** Characteristics of encephalitis patients positive for B19V DNA by sex.

Characteristic	Male N (%)	Female N (%)	Total N (%)
Patient	9 (69.2)	4 (30.8)	13 (100)
Age (Year)			
[0–5]	4 (80)	1(20)	5(100)
[6–18]	2 (50)	2 (50)	4 (100)
[19–40]	1 (50)	1 (50)	2 (100)
[41–60]	2 (100)		2 (100)
Mean	18.3 [2–53]	13 [5–27]	16.7 [2–53]
Fever			
Yes	6 (75)	2 (25)	8 (100)
No	2 (66.7)	1 (33.3)	3 (100)
Missing	1 (50)	1 (50)	2 (100)
Pleocytosis (nb/µL)			
[0–4]	6 (100)		6 (100)
[5–99]	0	2 (100)	2 (100)
Missing	3 (60)	2 (40)	5 (100)
% Lymphocytes			
[0–49]	0	2 (100)	2 (100)
[50–100]	0	0	0
Missing	3 (60)	2 (40)	5 (100)
Glycorrhachia (g/L)			
[0–0.3]	0	0	0
[0.4–0.8]	5 (71.4)	2 (28.6)	7 (100)
>0.8	1 (100)	0	1 (100)
Missing	3 (60)	2 (40)	5 (100)
Proteinorrachia (g/L)			
[0, 1]	6 (85.7)	1 (14.3)	7 (100)
>1		1 (100)	1 (100)
Missing	3 (60)	2 (40)	5 (100)

The proportion (%) of lymphocytes in the CSF was calculated only when the pleocytosis was equal to or greater than 5 elements/µL. N: number.

**Table 2 viruses-17-00111-t002:** Evaluation of the probable impact of novel mutations in B19V sequences obtained from cerebrospinal fluid samples in Senegal using SIFT software (https://sift.bii.a-star.edu.sg/, accessed on 2 December 2024).

Mutations	Protein	Protein Impact	Scores
p. Glu 114 Gly	NS1	Tolerated	0.13
**p. Ile 181 Met**	**NS1**	**APF**	**0.03**
p. Thr 182 Ala	NS1	Tolerated	0.47
p. Thr 279 Ala	NS1	Tolerated	0.16
p. Lys 312 Arg	NS1	Tolerated	0.63
**p. Val 30 Leu**	**VP1**	**APF**	**0.00**
**p. Ser 98 Asn**	**VP1**	**APF**	**0.00**
**p. His 100 Tyr**	**VP1**	**APF**	**0.00**
p. Thr 347 Ser	VP1; VP2	Tolerated	0.62
p. Asn 533 Ser	VP1; VP2	Tolerated	0.16

NS1: non-structural protein; VP1: capsid protein 1; VP2: capsid protein 2; APF: affect protein function. Bold represents mutations that affect protein function and their scores.

## Data Availability

The original contributions presented in the study are included in the article/Appendix A, further inquiries can be directed to the corresponding author/s.

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
