# Peer review of "Clinical Features of Human Parvovirus B19-Associated Encephalitis Identified in the Dakar Region, Senegal, and Viral Genome Characterization"

_viruses, 2025, doi:10.3390/v17010111_

Round 1
Reviewer 1 Report
Comments and Suggestions for Authors
This study reports clinical features of 13 patients with encephalitis, both children and adults from Senegal, with parvovirus B19 DNA detected by a RT-PCR panel (?) in cerebrospinal fluid. It also reports the full B19V-DNA sequences from CSF from three of these 13 patients. However, it does not reveal the other virus findings.
General concerns:
Even if the Intro is interesting, it is too long and includes much that could be deleted or moved to the Discussion. This is now written more like a review, which the Editors may comment on.
Further, the Title does not emphasize the clinical part, which again is highlighted in the Abstract, which is confusing, considering that sequence data was available only from 3 patients among which the genetic diversity was not very wide at all…
Moreover, this is not a description of a true “association”, because patients without encephalitis or with other brain diseases were not statistically compared.
This paper does not reveal how many of the total patients with encephalitis, were positive for B19V DNA in CSF. This would be highly interesting to know, also per age group. It does not reveal the other virus findings obtained with the same PCR panel either. Without that additional data, the results are quite thin, but if this data would be added, it would improve the quality of this paper, and make it worthy of this Journal.
Specific comments:
Authors:
Why are the surnames of some authors written in capital letters, while those of others are not?
Abstract:
Line 19: “…cases of humans infected with parvovirus B19-associated encephalitis”. Humans cannot be “infected with encephalitis”, please re-write.
Line 20: “patients mostly included children”. What is the age group defined as children? There is an age group of 16-40 years, so are patients below 16 years of age considered children, and patients above 16 years as adults? At least this reviewer considers children as <18 years. Please clarify your age grouping.
Line 23: Please open the abbreviation RT in RT-PCR. Was it “reverse transcription”? If so, why was RT-PCR used for a DNA virus? Please give a short clarification.
Also, lines 196-198: Why RNA denaturation and removal of RNA residues in B19V sequencing?
Lines 26-25: “B19 must be considered in differential diagnosis”. This sentence is confusing, do the authors mean in general/always, regardless of symptoms/signs/disease…? – because then follows “particularly in the presence of neurological symptoms”. It would be more clear to say “B19V must be considered in differential diagnosis of neurological symptoms”, if this is what is meant.
Introduction:
Lines 39-42: This sentence is very confusing,: “Whilst it resolves spontaneously without adverse outcomes, B19V infection occurs vertically from mother to fetus during pregnancy, leading to lysis of fetal red blood cells, fetal hydrops, spontaneous abortion and fetal death”. How can it resolve spontaneously when it leads to fetal death? Please re-write.
Line 52: The correct name for the genus is “Erythroparvovirus”, and all such genus and family taxon names should be written in italics (i.e., also Parvoviridae).
Line 53: “that binds to the P-cell receptor antigen on erythrocytes”. This is erroneous in two ways! The P-cell receptor is a hormone receptor in the pancreas and B19V does not infect mature erythrocytes. Perhaps the authors mean the blood-group P-antigen here, also called globoside? This has earlier been thought to be the main B19V receptor, but new research has changed this. Instead, B19V uses globoside only to transcytose the acid airway mucosa, while it uses another unknown VP1u-binding receptor in the neutral bone marrow to bind to the permissive erythrocyte precursors, although globoside is still needed for the virus to get out from the acidic intracellular endosomes to find their way into the nucleus. B19V does not bind, infect, or replicate in mature erythrocytes (red blood cells), which does not even have a nucleus.
Lines 61-63: About intragenotype distance values: “lower for genotype 1 (1–3%)” but later: “Genotypes 1 … subtypes a and b with the divergence of about 5%”. This can be misunderstood, since 5% is not within 1-3%? Please clarify.
Line 66, reference #8: It would be better to cite a newer review, because more info may have been gathered in the past 20 years...
Line 94: Please add all relevant references here after “encephalopathy”.
Line 96: “B19V infection is more common in immunocompromised hosts”. Is this really true? Please provide references if so. Ref 12 does not compare immunocompromised and immunocompetent, and they studied only neurological complications…? Do the authors mean neurological manifestations of B19V only? Please clarify.
Line 101: Please give a reference also for “steroid therapies.”
Lines 116-117: It is enough to open up an abbreviation once, the first time mentioned (VLP)
Line 138: “analysis of the CSF virus”. What virus is this? Or should it be “viruses in CSF”? Please clarify.
Line 140: “13 of 403 (3.2%) samples tested from meningoencephalitis cases in India”. Please include sample type, was it CSF?
Line 185: This real-time RT-PCR seems to be a panel of many different viruses, this should be mentioned. Please reveal also what other viruses the panel included.
Results
Line 249: “a total of 13 patients tested positive for B19V by RT-PCR” please add if this was in blood, throat, stools or CSF? Be specific, would also be best to say “B19V DNA”.
Line 253 and 269: “contained less than five elements per µL”. What elements?
Line 256: “(five data were missing for CSF biological and biochemical parameters)”. Do you mean "data of CSF biological and biochemical parameters were missing for five patients"?
Lines 257-262: These percentages do not make sense. How many patients of those 13 B19V-positive patients make 9.1%, 18.2% and 27.3%? (1/13 is 7.7%, 2/13 is 15.4%, 3/13 is 23.1%, 4/13 is 30.8%) Please clarify.
Line 258: Please clarify what “infectious syndrome” means in this context.
Line 263: “was coinfected with herpes simplex virus 1”. Please specify “coinfected” here (HSV1 DNA in CSF, latent infection/IgG positive, or cold sore?) or maybe "co-detected" would be better? How was this detected? Apparently, this RT-PCR is a panel with many viruses, right?
Please reveal what other viruses were screened and detected, and if there were other co-detections with B19V, or was this HSV1 the only one? If so, this should be told. How common was B19V then overall (by age group)?
Table 1: It would be more informative if the specific ages would be given when having this few patients, e.g., age group 16-40 is rather broad, are the 2 individuals 16 years (children), or close to 40 (adults)? Also, in the youngest age group, are they below 6 months of age, or 4-5 years?
Further: some brackets are wrong [[, and typo in Proteinorachie (Proteinorachia).
Line 273: In Table A of supplement there was no PQ390714, only these: PQ390712 and PQ390713 (and two with the same number PQ390712)? Please correct.
Line 297 Table 2: “Evaluation of the impact of novel mutations in B19V sequences obtained from cerebrospinal fluid samples in Senegal using SIFT software”? The SIFT software was not used to obtain samples, re-write: "Evaluation by SIFT software, of the impact of..."
Discussion
Lines 306-307: “B19V has been previously identified as responsible of encephalitis with a prevalence of 39% in humans, including 86% children”. This is not true when reading what it says! This must be among patients with encephalitis, not among humans and children? Please re-write.
Line 312: “with the 0-5 age group being more likely to be B19V-positive.” How many children with encephalitis of each patient group were there all together, and what was the percentage of B19V-DNA+ CSFs among those groups, as opposed to other etiologies? This would be interesting to mention and compare. It may be that children had more often encephalitis than older children and adults, thereby distorting the B19V percentages given here. Are they thus really "more likely to be B19V positive" or were there only so much more encephalitis cases in general among smaller children?
Moreover, please re-write this sentence, to include B19V-positive of what (DNA or IgM, IgG?) and where (in CSF?), and within which disease group (neurological symptoms or meningoencephalitis?), or instead of "B19V positive", say "caused by B19V" or "having acute B19V infection" or whatever the truth is...
Line 313-314: “our study is similar to that previously reported in USA”. Was this other study also about neurological complications or in general?
Further: Perhaps “B19V encephalitis and encephalopathy” should be “B19V encephalitis or encephalopathy”?
Line 316: “These data may suggest that children of all sexes are at greater risk of developing B19V-associated encephalitis.” Or do you mean “any encephalitis” or both any and B19V+?
Line 317: “B19 infects 1-5% of pregnant women”. Please give a reference.
Line 328: “more than 5 cells/m3”: Do you really mean cubic meter, or maybe cubic mm??
Author Response
Reviewer #1: Even if the Intro is interesting, it is too long and includes much that could be deleted or moved to the Discussion. This is now written more like a review, which the Editors may comment on.
Further, the Title does not emphasize the clinical part, which again is highlighted in the Abstract, which is confusing, considering that sequence data was available only from 3 patients among which the genetic diversity was not very wide at all…
Response: We thank the reviewer for these comments. These sections have been edited in the revised version of the manuscript.
Moreover, this is not a description of a true “association”, because patients without encephalitis or with other brain diseases were not statistically compared.
This paper does not reveal how many of the total patients with encephalitis, were positive for B19V DNA in CSF. This would be highly interesting to know, also per age group. It does not reveal the other virus findings obtained with the same PCR panel either. Without that additional data, the results are quite thin, but if this data would be added, it would improve the quality of this paper, and make it worthy of this Journal.
Response: We thank the reviewer for these comments. More details have been added to the revised version of the manuscript.
- Why are the surnames of some authors written in capital letters, while those of others are not?
Response: This was a typo and has been corrected in the revised version of the manuscript
- Line 19: “…cases of humans infected with parvovirus B19-associated encephalitis”. Humans cannot be “infected with encephalitis”, please re-write.
Response: This sentence has been reformulated in the revised version of the manuscript
- Line 20: “patients mostly included children”. What is the age group defined as children? There is an age group of 16-40 years, so are patients below 16 years of age considered children, and patients above 16 years as adults? At least this reviewer considers children as <18 years. Please clarify your age grouping.
Response: Age groups have been revised for greater clarity in the revised version of the manuscript.
- Line 23: Please open the abbreviation RT in RT-PCR. Was it “reverse transcription”? If so, why was RT-PCR used for a DNA virus? Please give a short clarification.
Also, lines 196-198: Why RNA denaturation and removal of RNA residues in B19V sequencing?
Response: We thank the reviewer for this comment. This section has been edited for more clarity in the revised version of the manuscript.
- Lines 26-25: “B19 must be considered in differential diagnosis”.This sentence is confusing, do the authors mean in general/always, regardless of symptoms/signs/disease…? – because then follows “particularly in the presence of neurological symptoms”. It would be more clear to say “B19V must be considered in differential diagnosis of neurological symptoms”, if this is what is meant.
Response: This has been corrected in the revised version of the manuscript
- Lines 39-42: This sentence is very confusing: “Whilst it resolves spontaneously without adverse outcomes, B19V infection occurs vertically from mother to fetus during pregnancy, leading to lysis of fetal red blood cells, fetal hydrops, spontaneous abortion and fetal death”. How can it resolve spontaneously when it leads to fetal death? Please re-write.
Response: This sentence has been rephrased in the revised version of the manuscript.
- Line 52: The correct name for the genus is “Erythroparvovirus”, and all such genus and family taxon names should be written in italics (i.e., alsoParvoviridae).
Response: This was a typo and has been corrected in the revised version of the manuscript
- Line 53: “that binds to the P-cell receptor antigen on erythrocytes”. This is erroneous in two ways! The P-cell receptor is a hormone receptor in the pancreas and B19V does not infect mature erythrocytes. Perhaps the authors mean the blood-group P-antigen here, also called globoside? This has earlier been thought to be the main B19V receptor, but new research has changed this. Instead, B19V uses globoside only to transcytose the acid airway mucosa, while it uses another unknown VP1u-binding receptor in the neutral bone marrow to bind to the permissive erythrocyte precursors, although globoside is still needed for the virus to get out from the acidic intracellular endosomes to find their way into the nucleus. B19V does not bind, infect, or replicate in mature erythrocytes (red blood cells), which does not even have a nucleus.
Response: This has been corrected in the final document with the support of a reference.
- Lines 61-63: About intragenotype distance values: “lower for genotype 1 (1–3%)” but later: “Genotypes 1 … subtypes a and b with the divergence of about 5%”. This can be misunderstood, since 5% is not within 1-3%? Please clarify.
Response: We thank the reviewer for this comment. The sentence have been edited for more clarity
- Line 66, reference #8: It would be better to cite a newer review, because more info may have been gathered in the past 20 years...
Response: “Manaresi, E., & Gallinella, G. (2019). Advances in the Development of Antiviral Strategies against Parvovirus B19. https://doi.org/10.3390/v11070659” was used instead of reference 4.
- Ligne 94: Please add all relevant references here after “encephalopathy”.
Response: References have been updated in the revised version of the manuscript
- Line 96: “B19V infection is more common in immunocompromised hosts”. Is this really true? Please provide references if so. Ref 12 does not compare immunocompromised and immunocompetent, and they studied only neurological complications…? Do the authors mean neurological manifestations of B19V only? Please clarify.
Response: Reference 12 has been replaced by a reference (https://doi.org/10.1128/CMR.15.3.485-505.2002) that best justifies the occurrence of B19V in immunocompetent and immunocompromised patients.
- Line 101: Please give a reference also for “steroid therapies.”
Response: A reference has been added
- Lines 116-117: It is enough to open up an abbreviation once, the first time mentioned (VLP)
Response: This has been corrected in the revised version of the manuscript
- Line 138: “analysis of the CSF virus”. What virus is this? Or should it be “viruses in CSF”? Please clarify.
Response: This has been corrected in the revised version of the manuscript
- Line 140: “13 of 403 (3.2%) samples tested from meningoencephalitis cases in India”. Please include sample type, was it CSF ?
Response: These are CSF samples. This has been included in the sentence
- Line 185: This real-time RT-PCR seems to be a panel of many different viruses, this should be mentioned. Please reveal also what other viruses the panel included.
Response: The other viruses included in the real-time PCR panel used have been listed in the revised version of the manuscript.
- Line 249: “a total of 13 patients tested positive for B19V by RT-PCR” please add if this was in blood, throat, stools or CSF? Be specific, would also be best to say “B19V DNA”.
Response: The type of sample used was specified in the revised version of the manuscript
- Line 253 and 269: “contained less than five elements per µL”. What elements ?
Response: Elements refer to leukocytes. This has been corrected in the sentence for more clarity.
- Line 256: “(five data were missing for CSF biological and biochemical parameters)”.Do you mean "data of CSF biological and biochemical parameters were missing for five patients"?
Response: Yes. This sentence has been rephrased in the revised version of the manuscript.
- Lines 257-262: These percentages do not make sense. How many patients of those 13 B19V-positive patients make 9.1%, 18.2% and 27.3%? (1/13 is 7.7%, 2/13 is 15.4%, 3/13 is 23.1%, 4/13 is 30.8%) Please clarify.
Response: At the end of this sentence, we added the number of missing data for more clarity.
- Line 258: Please clarify what “infectious syndrome” means in this context.
Response: The sentence has been rephrased for more clarity.
- Line 263: “was coinfected with herpes simplex virus 1”. Please specify “coinfected” here (HSV1 DNA in CSF, latent infection/IgG positive, or cold sore?) or maybe "co-detected" would be better? How was this detected? Apparently, this RT-PCR is a panel with many viruses, right?
Please reveal what other viruses were screened and detected, and if there were other co-detections with B19V, or was this HSV1 the only one? If so, this should be told. How common was B19V then overall (by age group)?
Response: The sentence has been rephrased for more clarity.
- Table 1: It would be more informative if the specific ages would be given when having these few patients, e.g., age group 16-40 is rather broad, are the 2 individuals 16 years (children), or close to 40 (adults)? Also, in the youngest age group, are they below 6 months of age, or 4-5 years?
Further: some brackets are wrong [[, and typo in Proteinorachie (Proteinorachia).
Response: Age groups have been revised for greater clarity in the revised version of the manuscript.
- Line 273: In Table A of supplement there was no PQ390714, only these: PQ390712 and PQ390713 (and two with the same number PQ390712)? Please correct.
Response: This was a typo and has been corrected in the revised version of the supplementary materials
- Ligne 297 Tableau 2 : « Evaluation de l'impact de nouvelles mutations dans les séquences B19V obtenues à partir d'échantillons de liquide céphalo-rachidien au Sénégal à l'aide du logiciel SIFT» ? Le logiciel SIFT n'a pas été utilisé pour obtenir des échantillons, réécrire : « Evaluation par le logiciel SIFT, de l'impact de... »
Response: This has been corrected in the revised version of the manuscript
- Lines 306-307: “B19V has been previously identified as responsible of encephalitis with a prevalence of 39% in humans, including 86% children”. This is not true when reading what it says! This must be among patients with encephalitis, not among humans and children? Please re-write.
Response: This has been rephrased in the revised version of the manuscript
- Line 312: “with the 0-5 age group being more likely to be B19V-positive.” How many children with encephalitis of each patient group were there all together, and what was the percentage of B19V-DNA+ CSFs among those groups, as opposed to other etiologies? This would be interesting to mention and compare. It may be that children had more often encephalitis than older children and adults, thereby distorting the B19V percentages given here. Are they thus really "more likely to be B19V positive" or were there only so much more encephalitis cases in general among smaller children?
Moreover, please re-write this sentence, to include B19V-positive of what (DNA or IgM, IgG?) and where (in CSF?), and within which disease group (neurological symptoms or meningoencephalitis?), or instead of "B19V positive", say "caused by B19V" or "having acute B19V infection" or whatever the truth is...
Response: This study does not compare the etiologies of encephalitis according to age group. But based on the 13 patients included in this study, there are more cases of viral encephalitis caused by B19V in children aged 0 to 5 years. This sentence has been rephrased in the revised version of the manuscript for more clarity.
- Line 313-314: “our study is similar to that previously reported in USA”. Was this other study also about neurological complications or in general?
Further: Perhaps “B19V encephalitis and encephalopathy” should be “B19V encephalitis or encephalopathy”?
Response: We thank the reviewer for this comment. The study in question refers to Neurologic Manifestations Associated with Parvovirus B19 Infection (https://doi.org/10.1086/599042). The sentence has been rephrased accordingly.
- Line 316: “These data may suggest that children of all sexes are at greater risk of developing B19V-associated encephalitis.” Or do you mean “any encephalitis” or both any and B19V+?
Response: We thank the reviewer for this comment. So we describe only data from B19V encephalitis as described in this study
- Line 317: “B19 infects 1-5% of pregnant women”. Please give a reference.
Response: A reference has been added
- Line 328: “more than 5 cells/m3”: Do you really mean cubic meter, or maybe cubic mm??
Response: It’s cubic mm, this was a typo and has been corrected in the revised version of the manuscript
Reviewer 2 Report
Comments and Suggestions for Authors
Some suggestions for improvement:
· Some sentences in the introduction are lengthy and could be simplified for better readability
· The authors should provide more details about the patient recruitment process and inclusion/exclusion criteria
· The statistical analysis section could be expanded to better explain how the genetic diversity parameters were calculated
· Consider including power calculations to justify the sample size
· Table 1 could benefit from statistical analyses (p-values) to strengthen the observations regarding age and gender distributions
· The phylogenetic tree (Figure 1) is of poor quality and would be more informative with additional details about the bootstrap values and scale bar
· Consider adding a figure showing the location of the novel mutations identified in the viral genome
· The implications of the identified mutations could be discussed in more depth, particularly regarding their potential impact on viral pathogenesis
· The limitations of the study should be more explicitly stated.
The authors should:
1. Expand the methods section with requested details
2. Improve the quality of the presentation of results, as well as with additional statistical analyses
3. Strengthen the discussion by addressing the suggested points
4. Address the minor editorial issues.Recommendation: I recommend this manuscript for publication after minor revisions. The suggested improvements would enhance the clarity and impact of this work.

Author Response
Reviewer #2: Some suggestions for improvement:
- Some sentences in the introduction are lengthy and could be simplified for better readability
- The authors should provide more details about the patient recruitment process and inclusion/exclusion criteria
- The statistical analysis section could be expanded to better explain how the genetic diversity parameters were calculated
- Consider including power calculations to justify the sample size
- Table 1 could benefit from statistical analyses (p-values) to strengthen the observations regarding age and gender distributions
- The phylogenetic tree (Figure 1) is of poor quality and would be more informative with additional details about the bootstrap values and scale bar
- Consider adding a figure showing the location of the novel mutations identified in the viral genome
- The implications of the identified mutations could be discussed in more depth, particularly regarding their potential impact on viral pathogenesis
- The limitations of the study should be more explicitly stated.
The authors should:
- Expand the methods section with requested details
Response: We thank the reviewer for these comments. These sections have been edited and more details have been added in the revised version of the manuscript
- Improve the quality of the presentation of results, as well as with additional statistical analyses
Response: The phylogenetic tree quality has been improved in the revised version of the manuscript. Given the small number of samples included in this study (n=13), statistical tests are not possible.
- Strengthen the discussion by addressing the suggested points
Response: The discussion was edited and improved in the revised version of the manuscript
- Address the minor editorial issues. Recommendation: I recommend this manuscript for publication after minor revisions. The suggested improvements would enhance the clarity and impact of this work.
Response: We thank the reviewer for this recommendation and his comments to enhance the clarity of our work.
Round 2
Reviewer 1 Report
Comments and Suggestions for Authors
Previous comments have been taken into account. However, please correct these:
Abstract, lines 21-23: Spell out CSF (also on line 82), and re-write “in 0-5 years children”
Line 56: there is no ref 68, also ref 6 is inadequate here, delete! You can cite e.g., this instead: Bircher et al. Viruses, 2022 (PMID: 35216013).
Lines 67-68: “between 1-3% for the genotype 1 (1–3%)”: Delete the parenthesis.
Lines 107-109: A 15-year-old person cannot be both a child and an adult at the same time. Change either ≤ or ≥ 15 years.
Lines 131 and 135: The “RNA extracts” and the “DNA extracts” should be the other way around!
Lines 196-197: “Etiologies included herpesviruses, enteroviruses, respiratory viruses have been found.” Delete “have been found” or change “included” to “including”.
Line 260: “B19V-associated encephalitis is rare but can occur in humans, particularly in children, even in the absence of typical symptoms.” This means even without typical symptoms of encephalitis. Please change to “B19V-typical symptoms”.
Line 279: “B19V infects 1-5% of pregnant women, usually with a normal pregnancy [22].” Ref 22 seems to be inappropriate, an epidemiological pregnancy study would be better.
Line 289: “(25%, 2/8)”, should this not be “(15.4%, 2/13)”, as stated in the beginning of the sentence?
Line 332: Perhaps the paper Bircher et al. (above) could be of interest here too regarding VP1u interactions with the cell?
Author Response
Response to Editor and Reviewer comments:  
The revised manuscript ' Genetic diversity of Human Parvovirus B19 associated with viral encephalitis in the Dakar region, Senegal' has addressed the majority of the reviewers comments. The following questions remain on this version of the manuscript:
Reviewer: Previous comments have been taken into account. However, please correct these:
- Abstract, lines 21-23: Spell out CSF (also on line 82), and re-write “in 0-5 years children”
Response: We thank the reviewer for this comment. This was a typo and has been corrected in the revised version of the manuscript
- Line 56: there is no ref 68, also ref 6 is inadequate here, delete! You can cite e.g., this instead: Bircher et al. Viruses, 2022 (PMID: 35216013).
Response: This was a typo and has been corrected in the revised version of the manuscript. Reference 6 was also changed.
- Lines 67-68: “between 1-3% for the genotype 1 (1–3%)”: Delete the parenthesis.
Response: This was a typo and has been corrected in the revised version of the manuscript.
- Lines 107-109: A 15-year-old person cannot be both a child and an adult at the same time. Change either ≤ or ≥ 15 years.
Response: This was a typo and has been corrected in the revised version of the manuscript.
- Lines 131 and 135: The “RNA extracts” and the “DNA extracts” should be the other way around!
Response: This was a typo and has been corrected in the revised version of the manuscript.
- Lines 196-197: “Etiologies included herpesviruses, enteroviruses, respiratory viruses have been found.” Delete “have been found” or change “included” to “including”.
Response: This has been corrected in the revised version of the manuscript.
- Line 260: “B19V-associated encephalitis is rare but can occur in humans, particularly in children, even in the absence of typical symptoms.” This means even without typical symptoms of encephalitis. Please change to “B19V-typical symptoms”.
Response: This has been corrected in the revised version of the manuscript.
- Line 279: “B19V infects 1-5% of pregnant women, usually with a normal pregnancy [22].” Ref 22 seems to be inappropriate, an epidemiological pregnancy study would be better.
Response: A reference has been added to this section in the revised version of the manuscript. The sentence has been rephrased according to the reviewer’s suggestion.
- Line 289: “(25%, 2/8)”, should this not be “(15.4%, 2/13)”, as stated in the beginning of the sentence?
Response: missing data for pleocytosis were recorded in 5 patients. Only 8 patients had pleocytosis results.
- Line 332: Perhaps the paper Bircher et al. (above) could be of interest here too regarding VP1u interactions with the cell?
Response: This section has been improved in the revised version of the manuscript.